# Transposable Elements Shape the Genome Diversity and the Evolution of Noctuidae Species

**DOI:** 10.3390/genes14061244

**Published:** 2023-06-10

**Authors:** Chunhui Zhang, Lei Wang, Liang Dou, Bisong Yue, Jinchuan Xing, Jing Li

**Affiliations:** 1Key Laboratory of Bio-Resources and Eco-Environment (Ministry of Education), College of Life Sciences, Sichuan University, Chengdu 610065, China; 2Department of Genetics, Human Genetics Institute of New Jersey, Rutgers, State University of New Jersey, Piscataway, NJ 08854, USA

**Keywords:** Noctuidae, transposable elements, genomic diversity, phylogeny, horizontal transfer TE (HTT) events

## Abstract

Noctuidae is known to have high species diversity, although the genomic diversity of Noctuidae species has yet to be studied extensively. Investigation of transposable elements (TEs) in this family can improve our understanding of the genomic diversity of Noctuidae. In this study, we annotated and characterized genome-wide TEs in ten noctuid species belonging to seven genera. With multiple annotation pipelines, we constructed a consensus sequence library containing 1038–2826 TE consensus. The genome content of TEs showed high variation in the ten Noctuidae genomes, ranging from 11.3% to 45.0%. The relatedness analysis indicated that the TE content, especially the content of LINEs and DNA transposons, is positively correlated with the genome size (r = 0.86, *p*-value = 0.001). We identified SINE/*B2* as a lineage-specific subfamily in *Trichoplusia ni,* a species-specific expansion of the LTR/*Gypsy* subfamily in *Spodoptera exigua*, and a recent expansion of SINE/*5S* subfamily in *Busseola fusca.* We further revealed that of the four TE classes, only LINEs showed phylogenetic signals with high confidence. We also examined how the expansion of TEs contributed to the evolution of noctuid genomes. Moreover, we identified 56 horizontal transfer TE (HTT) events among the ten noctuid species and at least three HTT events between the nine Noctuidae species and 11 non-noctuid arthropods. One of the HTT events of a *Gypsy* transposon might have caused the recent expansion of the *Gypsy* subfamily in the *S. exigua* genome. By determining the TE content, dynamics, and HTT events in the Noctuidae genomes, our study emphasized that TE activities and HTT events substantially impacted the Noctuidae genome evolution.

## 1. Introduction

The family Noctuidae is highly diverse in species, with almost 12,000 species, forming the third largest family within the order Lepidoptera [1]. Many of this family are phytophagous insects and highly harmful to crops or forests. Despite the large number of previous studies focusing on the morphology, physiology, and biological control of Noctuidae species [2,3], our understanding of the genomic diversity of the Noctuidae species, especially the transposable elements (TEs), is still in its infancy.

TEs are a class of repetitive sequences dispersed throughout the genome. They are essential components of eukaryotic genomes that can mobilize around the genome within the same chromosome or between different chromosomes, even transfer horizontally between species [4,5]. TEs move through the genomes using either a “cut-and-paste” or a “copy-and-paste” mechanism, and TEs have important impacts on the architecture, function, and evolution of the host genome [6,7,8].

In recent decades, many insect genomes have been assembled. Insect genomes vary significantly in size, ranging from as large as 6.5 Gb in *Locusta migratoria* to as small as less than 0.1 Gb in *Tetranychus urticae* [9,10]. However, the number of genes in their genomes is similar, and the difference in genome size is mainly due to variations in TE contents [11]. Previous studies on Arthropoda, Lepidoptera, and the genus *Drosophila* suggested a positive correlation of genome size to the TE content [7,11,12]. Nevertheless, it is especially challenging to study insect TEs for several reasons. (1) Insect genomes vary greatly in size and the proportion of TEs. For example, two species in the order Diptera, *Aedes aegypti* and *Belgica Antarctica*, have a TE genome content of 55% and 1%, respectively. Even in the same genus, *Drosophila simulans* and *D. ananassae* have significant differences in TE content, at 10% and 40% of the genome, respectively [13]. (2) Many lineage-specific TEs exist in different insect genomes, such as the *Zisupton* subfamily specific to coleopterans genomes [13]. (3) The TE composition is also highly variable in insect genomes. For example, DNA transposons are the predominant TE class in *Heliconius melpomene,* a species in the order Lepidoptera, whereas DNA transposons have a very low genome proportion in *Papilio polytes*, another species in the order Lepidoptera [7]. (4) TE propagation is significantly different in insect genomes. Wu’s study found that out of 14 arthropod species, only silkworms had a large number of recent expansion TEs, which probably was responsible for the adaptation to domestication in the silkworms [11].

In addition to the large TE variation, another challenge in studying insect TEs is the existence of horizontal transfer TE (HTT) events in insect genomes. TEs can transfer from one host to another in two ways. The first is vertical inheritance, where they are passed from parents to offspring. The second way is HTT, which occurs between organisms that do not mate [14]. The HTT allows TEs to jump from one host to another. In the old host genome, natural selection and silencing mechanisms can suppress the propagation or delete TEs from the genome. However, when it is inserted into a new host genome, it can escape the suppression and extinction [15]. Therefore, HTT plays an important role in the long-term survival of TEs. Since the first HTT event was reported in *D. melanogaster* [16], a total of 2836 HTT events have been recorded in HTT-db by 2017 [17]. One of the most recent HTT events occurred after 2010 in *D. simulans*. A *P* element horizontally transferred from the *D. melanogaster* into the *D. simulans* genome, and the *P* element could be found in the populations of *D. simulans* only after 2010 [18]. Previous studies suggested that order Lepidoptera is a hotspot for HTT events [19]. As one of the largest families of Noctuidae in the order Lepidoptera, there is no comprehensive study of HTT events among Noctuidae species, as well as between Noctuidae species and non-noctuid arthropods to date.

By 2020, genome assemblies are available for ten species of the Noctuidae. The ten species belong to seven genera. With multiple prediction methods, in this study, we annotated and characterized TEs in the genomes of the ten species to reveal the genomic diversity of TEs in Noctuidae and the correlation of TE content to the genome size of noctuids. We also investigated how different TE classes/subfamilies expanded/contracted in the genome of Noctuidae insects. We also estimated HTT events among the genome of Noctuidae species, as well as between Noctuidae species and other arthropods, and elucidated how HTT events affect the evolution of the Noctuidae genome.

## 2. Materials and Methods

### 2.1. Data Collection

Ten species of Noctuidae belonging to seven genera with published genomes were selected (Appendix A), including *Agrotis ipsilon*, *B. fusca*, *Helicoverpa armigera*, *Helic. zea*, *Heliothis virescens*, *Mamestra configurata*, *S. exigua*, *S. frugiperda*, *S. litura*, and *T. ni*. Two species are in the genus Helicoverpa, and three are in the Spodoptera genus. The genome sequences were downloaded from NCBI (https://www.ncbi.nlm.nih.gov/) (accessed on 30 September 2020). Published DNA and protein sequences of repetitive elements were downloaded from Repbase [20]. NR database was downloaded from the NCBI database. The TimeTree [21] was used to derive phylogenetic relationships of the Noctuidae species in conjunction with the literature, and iTOL [22] was used to generate the phylogenetic tree (Figure 1). Due to the unknown phylogenetic relationship of *Helio. virescens* [23,24], it was not included in the phylogenetic tree of Figure 1. Genome sequences and TE libraries of 11 non-noctuid arthropods were downloaded from ArTEdb (http://artedb.net/index.html) (accessed on 30 September 2020).

### 2.2. Species-Specific TE Library Construction

We used RepeatModeler 2.0 [25] to build consensus sequence libraries for each Noctuidae species. Since version 2.0, RepeatModeleris able to call tools such as LTR_harvest [26] and LTR_retriever [27] to make consensus sequences based on structure, in addition to calling RepeatScout [28] and RECON [29] to create consensus sequences based on repetitive sequence properties (-pa 12 -LTRStruct). Unknown types accounted for the majority of the results output by RepeatModeler2. We, therefore, attempted to classify the unknown types of consensus sequences by PASTEClassifier [30] and TEclass [31]. Consensus sequences were aligned to TE Repbase proteins using BlastX using default parameters. Only the aligned query sequence is kept. To remove potential protein-coding sequences that are not TE-encoded proteins, other sequences were aligned to Swiss–Prot using BlastX (identity > 30%, e-value < 1 × 10^−5^, percent query coverage > 50%), and the aligned sequences were excluded from the library. To filter non-coding RNAs such as tRNA and rRNA, an ncRNA library was created using tRNAscan-SE [32] and Rfam [33]. The consensus sequences were used as queries for Blastn to search against the ncRNA library using default parameters. To obtain the final TE libraries, the aligned query sequences were excluded.

### 2.3. TE Annotation and Statistical Analysis

Genomes were masked by RepeatMasker version 4.0.9 (http://repeatmasker.org/), and the “-lib” parameter was applied to use the custom TE library. We extracted TE copy sequences using the script ONE_CODE_TO_FIND_THEM_ALL.PL [34] with the “—strict” parameter and calculated the number of copies per family for each species via the self-definition function in Python. Pearson correlation analysis was applied via the “scipy.stats.pearsonr” function [35] in Python to analyze the correlation between genome size and TE load.

### 2.4. TE Propagation Activity

To estimate TE propagation activity during the evolutionary history of Noctuidae, we performed a copy-divergence analysis of the TE subfamilies based on their Kimura 2-parameter distances. The Kimura 2-Parameter divergence of TEs was calculated using buildSummary.pl and calcDivergenceFromAlign.pl on alignment files. Kimura distances were transformed to the time estimates of variation in TE activity with the following equation: T = K/2r, where r is the neutral mutation rate estimates, and K is the Kimura 2-Parameter divergence of TEs. The neutral mutation rate was set to 2.9 × 10^−9^ from *Heliconius melpomene* and assumed one generation per year in general [36].

### 2.5. Calculation of Phylogenetic Signal

Two tests were applied for phylogenetic signal analysis using the TE load. The first method was Pagel’s lambda. The value of lambda ranges from 0 to 1. The higher the value was, the stronger the phylogenetic signal was, indicating that the trait is highly correlated with phylogeny and not random. Another test was Blomberg’s K which quantifies the variance of traits relative to what we would expect under Brownian motion (BM). The value of K ranges from 0 to infinity. K = 0 means that there was no phylogenetic signal in the continuous trait. K = 1 meant that the trait had evolved under BM. K > 1 meant that there was more phylogenetic signal than expected under a BM model of trait evolution, which indicated that closely related species shared high similarities in traits. We applied phylogenetic signal analysis via the “phytools.phylosig” function in R [37]. According to the phylosig’s documentation, lambda value is influenced by the relative height in this function. Therefore, the calculated value may exceed 1.

### 2.6. Ancestral Node Reconstruction and Predicting the Change Rate of TEs

The phylogenetic signals of TE estimated by Pagel’s lambda method were used to test which standard phylogenetic comparative model was appropriate for ancestral state reconstruction. We used the “Geiger.fitContinuous” function [38] in R to evaluate the value of AICc in each of the following four standard phylogenetic comparative models: Brownian motion (BM), Ornstein–Uhlenbeck (OU), Early-burst (EB), and white noise. We compared the AICc values of the four models, and the results showed that the EB model was the most appropriate (Appendix A). The EB model was used for the ancestral state reconstruction of TE Loads. Based on the maximum likelihood method, the TE content of the ancestor nodes was reconstructed using the anc.ML function of the R package phytools (models = “EB”).

BAMM (Bayesian analysis of macroevolutionary mixtures) is a program for modeling trait evolution in the time-calibrated phylogeny [39,40,41]. We used the program to calculate evolutionary rates for TE loads in a phylogenetic tree. The “BAMMtools.setBAMMpriors” function was applied to estimate the betaInitPrior and betaShiftPrior for BAMM settings. The BAMM outputs were analyzed and visualized by BAMMtools [42].

### 2.7. Identification of TE Horizontal Transfer Events between Noctuidae Species

If some TE pairs showed more similarity than their hosts diverge, we suspected that an HTT event occurred [15]. We adopted the *dS* (synonymous substitutions per synonymous site) to identify HTT events. *dS* cannot be evaluated on the TE class that lacks proteins. Therefore, SINEs were excluded from our analysis. Using ONE_CODE_TO_FIND_THEM_ALL.PL, we extracted the autonomous TE sequences, which were greater than or equal to 80% of the length of the corresponding consensus sequences. TE copies were aligned to the TE protein library extracted from Repbase using tblasn. The best hits were realigned using exonerate, and the proteins of TE copies were extracted. All TE proteins were aligned between every pair of noctuid-selected species using blastp. TE pairs aligned over 300 bp were kept for calculation of *dS* using KaKs_calculator [43]. Based on the core genes of Lepidoptera, the BUSCO [44] pipeline was used to locate single-copy orthologous genes of the ten species.

Encoded peptide sequences of single-copy genes were aligned between every two noctuids using blastp (10^−4^). Proteins with reciprocal best hits and the same BUSCO identifier were identified as orthologous sequences. We only kept alignments > 300 amino acids and orthologs to calculate *dS* using KaKs_calculator. We define an HTT event as a TE pair with a *dS* smaller than the top ~2.5% of the total orthologous gene pairs.

### 2.8. Identification of TE Horizontal Transfer Events between Noctuidae Species and Other Arthropods

The basic principle was to compare TE pairs and host divergence rates. Searching for HTT events only between distant lineages could reduce the misclassification of HTT because the chance of having similar TE pairs inherited vertically in the distant lineage is very low. We divided 9 noctuids and 11 arthropod species into different lineages, and we only considered HTTs between lineages. Due to the unclear evolutionary relationship of *Helio. virescens*, the analysis was conducted on nine species of nocturnal moth insects instead of ten. Genome and TE libraries of 11 arthropods were downloaded from ArTEdb. We collapsed a clade of species into a lineage if (i) a fraction (>0.3%) [41] of its core orthologous genes showed lower *dS* than the highest nucleotide divergence of TEs or (ii) these species diverged in the last 40 My [45]. The study here referred to Peccoud’s heuristic search approach [45], and the clustering code was written based on a reference from Zhang Huahao’s vertebrate HTT study [46].

## 3. Results

### 3.1. Construction of TE Libraries in the Ten Noctuidae Species

We constructed a repetitive sequence library using RepeatModeler, which contained 1066–2904 consensus sequences for each of the ten different species (https://github.com/BonesAQ/te_families). Initially, we compared the library to protein sequence databases and removed non-transposable element (TE) encoding proteins, resulting in 1050–2835 consensus sequences from the ten species. We then filtered out RNA and simple repeats, leaving a library with 1038–2826 consensus sequences from the 10 species, including 241–603 retrotransposons (RTEs) and 172–350 DNA transposons (DTEs). Most of the consensus sequences were unclassified. Next, we used prediction and classification software, such as TEclass, to classify the unclassified sequences and found that the range of retrotransposon consensus in these 10 species was 562–1617, while that of DNA transposons was 428–1186 (Table 1 and Appendix A). In general, the bigger the genome size, the more consensus sequences were obtained. Genome sizes of the ten species varied from 299.98 Mb to 559.39 Mb, with the biggest genome in *M. configurata* and the smallest genome in *Helic. armigera.* Despite the success of the TEclass in classification, 3.08–5.26% of consensus sequences failed to be classified and were labeled as unknown repetitive sequences.

### 3.2. TE Contents

Genome content of TE showed high variations among Noctuidae genomes, accounting for 11.3–45.1% of the ten genomes (Table 2). Three species (*Helic. armigera*, *Helic. zea*, and *T. ni*) with the smallest genome sizes showed TE contents lower than 20%. *B. fusca* had the highest TE content (45.1%) and the second largest genome (490 Mb). In general, the bigger the genome was, the higher the TE content.

Four TE classes were annotated while their content varied greatly between ten noctuid genomes (Table 2). LINEs and DNA transposons were dominant, accounting for 3.46–20.16% and 4.01–12.1% of the entire genomes, respectively. SINEs and LTR retrotransposons accounted for 0.98–6.11% and 0.74–4.22% of the genomes, respectively. Notably, we found a large expansion of LINEs (20.16% of the genome) and DNA transposons (12.1% of the genome) in the *B. fusca* genome compared to other noctuids, which might contribute to the highest TE content (45.1%) in the *B. fusca* genome. In addition, an expansion of LTR occurred in *S. exigua* genome with the highest LTR percentage (4.22%) among the ten species, while the total TE content of *S. exigua* was only 30.64%. SINE showed reduced activity in *S. frugiperda* compared to other species. *S. frugiperda* genome had the lowest SINE percentage (0.98%), which was only 16–47% of that in other noctuid species. Radar plots in Appendix A showed the differences in TE proportions of noctuid species compared with *Helic. armigera* (lowest total TE content, 11.33%) and *B. fusca* (highest total TE content, 45.1%).

### 3.3. TE Subfamilies and Copy Numbers

A total of 40 subfamilies were identified in the ten Noctuidae species, including 13 LINE subfamilies, 4 SINE subfamilies, 6 LTR subfamilies, and 17 DNA subfamilies. A heat map of the 40 subfamilies across the ten species was plotted (Figure 2). The DNA/*Helitron* subfamily showed high copy numbers consistently in all ten species, with an average copy number of 141,201 (100,498–236,722). In contrast, copy numbers of other subfamilies varied greatly across species. For example, the SINE/5S subfamily had only 130–204 copies in three species of *Spodoptera* compared with relatively high abundance (>10,000 copies) in other species. Meanwhile, LINE/*R1* and LINE/*CR1* subfamily were abundant in copy number in most noctuids except three species (*Helic. armigera*, *Helic. zea*, and *T. ni*). The three species had much fewer copies of the *R1* subfamily (13,008–14,149 copies) and *CR1* subfamily (1188–1727 copies) than those of the other seven species (83,107–152,596 copies in *R1* and > 100,000 in *CR1*). We also found several lineage-specific subfamilies. The SINE/*B2* subfamily was only found in the *T. ni* genome with a high copy number (4784 copies). The DNA/*Crypton* subfamily was present specifically in *Helic. zea* (753 copies), *A. ipsilon* (37 copies), and *M. configurata* (1257 copies) genomes.

### 3.4. Propagation Activity of TE Subfamilies

Following the methods [7,47], we estimated the propagation activity of different TE subfamilies. We converted the 10% divergence rate to 17.24 million years based on the estimated neutral mutation rate. Figure 3 showed several subfamilies with distinct propagation activity among the ten species. Given that the SINE/*B2* subfamily was specific in *T. ni*, we found that SINE/*B2* began to insert into the *T. ni* genome around 60 million years ago (Mya) with a propagation peak of 20–30 Mya (Figure 3A). According to the phylogenetic tree (Figure 1), *T. ni* was a basal species diverged from nine other species, about 60 Mya. This indicated that SINE/*B2 was* inserted into *T. ni* after the divergence of the other species and resulted in the lineage-specific subfamily. LINE/*CR1* showed continuous propagation in most noctuids except three species (*Helic. armigera*, *Helic. Zea,* and *T. ni*), consistent with the high copy number of the *CR1*subfamily in many species. However, the timing of *CR1*propagation varied in different genomes. A burst propagation of *CR1* subfamily occurred 30–40 Mya in *S. exigua*, *and* 15–20 Mya in *S. frugiperda*, in contrast to very recently around 6 Mya in the *B. fusca* genome (Figure 3B). Figure 3C,D showed the insertion time distribution of LINE/*R1* and SINE/*5S* subfamily in the ten species. Interestingly, both subfamilies had a peak propagation around 6 Mya in the *B. fusca,* leading them to be the two most abundant TE subfamilies among the ten genomes. The DNA/*Maverick* subfamily was present only in five species, with the highest copy numbers in *S. frugiperda.*
Figure 3E showed an obvious propagation peak around 40 Mya in *S. frugiperda*.

Next, we compared the activity of the TE subfamily among three closely related species in the genus *Spodoptera*. Most TE subfamilies showed similar propagation activity among the three species, except LTR/*Gypsy* (Figure 3F). *The Gypsy* subfamily showed high activity in the *S. exigua* genome recently with a propagation peak of about 2 Mya, accumulating 7393 copies of *Gypsy* and a genome content of 2.25% in the *S. exigua* genome, which was 4.5 and 6.4 times that in *S. frugiperda* and *S. litura* genomes, respectively. Since *S. exigua* diverged from the other two species about 25 Mya (Figure 1), it was clear that the expansion of the *Gypsy* subfamily was a species-specific event.

### 3.5. Relatedness of TE Content with Genome Size of Noctuid Species

The TE content and the genome size of the ten noctuid species showed a strong positive correlation (r = 0.86, *p*-value = 0.001). We investigated which class or superfamily of TEs contributed more to the genome size variation. We listed all subfamilies with r values > 0.2 in Table 3. Both LINEs and DNA transposons showed a strong positive correlation with genome size with r > 0.8 and *p*-values < 0.005. While LTR and SINEs had moderate or weak correlations with genome size, these correlations are insignificant (*p*-values > 0.1).

Concerning subfamilies, DNA/*TcMar* and DNA/*Zator* subfamilies showed the strongest correlation with genome size (r > 0.7 and *p* < 0.01). Interestingly, the most abundant DNA transposons, the *Helitron* subfamily, had little correlation with genome size (r = 0.13). Within LINEs, *Dong, L2, and R1* subfamilies showed strong correlations with genome size (r > 0.7 and *p* < 0.01), while *RTE* and *CR1* had moderate correlations with genome size. All these subfamilies except *Dong* had high copy numbers in the noctuids genome.

### 3.6. Phylogenetic Signals and Changes of TE Load

We calculated lambda and K values to evaluate the phylogenetic signals in the four TE classes (Table 4). Although DNA transposon, LINE, and LTR had lambda values > 0.5, only LINE had a *p*-value < 0.05, indicating that LINE elements were correlated with phylogeny with high confidence. LINE also had a K value > 1, indicating that closely related species shared high similarity in the LINE content. A total of 12 subfamilies had lambda values > 0.5, six of which had *p*-values < 0.05 and K-values > 1 (*p* < 0.05), including DNA/hAT, DNA/TcMar, LINE/CR1, LINE/L2, LINE/RTE, and LINE/R1 (Table 4). This result suggests that the activity of these subfamilies is highly correlated with the phylogeny.

By predicting TE load and TE change rate on ancestral nodes of the phylogenetic tree, we estimated whether the TE expansion activity was associated with the noctuids’ phylogeny. SINE retrotransposons were excluded from the analysis, given that they contained very little phylogenetic signal (Table 4). Four comparative models were evaluated, and the EB model was selected (Appendix A). The maximum likelihood method was used to infer the TE loads in ancestor nodes (internal nodes). The black and red lines on the phylogenetic tree in Figure 4 represented expanded or reduced TE load in the genomes, respectively. An expansion was found in *B. fusca* in three classes of TE (LTR, LINE, and DNA). In contrast, all three classes of TE loads were reduced in *Helicoverpa* lineage. In *T. ni* lineage, an expansion was found in LTR, and a reduction in both DNA transposons and LINE. In *M. configurata*/*A. ipsilon* lineage, an expansion of LINE was observed. In the *Spodoptera* lineage, LTR and LINE expanded, compared to reduced activity of DNA transposons (Figure 4A–C). The TE expansion activity also varied among species within the same genus. For example, LTR and LINE activity reduced, but DNA transposon expanded in *S. frugiperda* which was different from the other two *Spodoptera* species.

Next, we compared the expansion activity of four LINE subfamilies (*CR1*, *L2*, *R1*, and *RTE*) across the nine noctuid species (Figure 4D–G). They were generally similar to each other and consistent with the results of LINE as a whole. However, we noted a reduction in *CR1* load (Figure 4D) and *R1* load (Figure 4F) in *M. configurata,* and a reduction in *CR1* (Figure 4D) and *L2* (Figure 4E) in *A. ipsilon*, while the overall LINE expanded in both species (Figure 4B). *RTE* subfamily in *S. exigua* showed a reduced activity (Figure 4G) compared to the expansion of LINE and other LINE subfamilies (*CR1*, *L2*, and *R1*). The expansion activity of two DNA subfamilies also was different across the noctuids. There was an expansion in hAT and overall DNA but a reduction in *TcMar* subfamily in *B. fusca* (Figure 4H,I). In addition, they also showed different expansion activity between the two *Helicoverpa* species.

We further applied BAMM to predict the change rate of TEs (Table 5). LINE had the fastest change rate, followed by DNA transposons, and LTR had a change rate much lower than LINE and DNA.

We next mapped change rates of TE on the phylogenetic tree to demonstrate the expansion activity changed over time (Figure 5). The warmer the color is, the higher the change rate was, and the nearer the color to the species name, it occurred more recently. The evolution of LTR, LINE, and DNA transposons (Figure 5A–C) was faster in ancient times than in recent times. While recently, LTR in *S. exigua,* LINE in *B. fusca,* and DNA transposons in *T. ni* showed the fastest change rate among the Noctuidae species. Within LINE subfamilies, change rates varied among different lineages. *CR1*subfamily in *S. exigua*, *B. fusca,* and *T. ni* evolved more rapidly than other species (Figure 5D). While the *R1* subfamily in *M. configurata* and *T. ni*, *the RTE* subfamily in *A. ipsilon* changed rapidly in recent times (Figure 5E–G). Within DNA subfamilies, the lineage *M. configurata*/*A. ipsilon* showed distinct change rates between the *hAT* and *TcMar* subfamily, and the evolution rate of the *hAT* subfamily was relatively fast in *M. configurata*/*A. ipsilon* compared with other species (Figure 5H,I).

### 3.7. Transposon Horizontal Transfer Events among Noctuidae Species

We identified millions of TE copies based on the constructed TE consensus sequences in the genomes of ten noctuid species. This dataset allowed us to examine the HTT events in these species. To do so, we obtained the single-copy genes in each species and their single-copy homologous genes in other species. *dS* values in all TEs and the homologous genes were calculated and compared to identify potential HTT events. A strict threshold was adopted in our analysis: a pair of TEs was considered to be an HTT event if their *dS* values are less than the *dS* value of the 2.5% of the orthologous genes with the lowest *dS* values in the two species (Appendix A). A total of 56 possible HTT events were identified, including 22 DNA transposons, 32 LINEs, and 2 LTR retrotransposons (Figure 6, Appendix A). Due to the unclear evolutionary relationship of *Helio. virescens*, the results related to *Helio. virescens* are not shown in the figure. However, the analysis in this section does not require a clear evolutionary relationship, so the total number of horizontal gene transfers still includes *Helio. virescens*. *S. exigua* was involved in 27 HTT events which was the highest among the ten species. We noted that both HTT events involving LTRs occurred in the *S. exigua* genome. One was LTR/*Copia* element transferring between the *S. exigua* and *T. ni* genome, another one was LTR/*Gyspy* element transferring between the *S. exigua* and the *M. configurata* genome (Appendix A). The results were consistent with the above findings that LTR expansion in *S. exigua* (Figure 4A) and the higher change rate recently in *S. exigua* (Figure 5A).

Due to misclassification, TEs from high-copy TE subfamilies were more likely to be identified in HTT events. To evaluate the potential bias, we calculated the number of HTT events per thousand copies in each TE family (Table 6). The DNA/*TcMar* and LINE/*RTE* subfamilies were involved in the greatest number of HTT events; however, neither of them had a high copy number. In contrast, DNA/*Maverick* had the highest frequency of HTT per thousand copies (1.72) among the subfamilies but the lowest copy number. While DNA/Helitron had the lowest frequency of HTT (0.001) per copy with the highest copy number among all subfamilies. The results indicated that subfamilies with high copy numbers were not correlated with more HTT events.

### 3.8. Transposon Horizontal Transfer Events between 9 Noctuidae Species and Other Arthropods

We further identified HTT events between Noctuidae species and other arthropods. TE libraries of 11 non-Noctuidae arthropods were obtained from the ArTEdb [11] and combined with TEs from nine Noctuidae species for HTT analysis. As stated in the method, this section requires a clear evolutionary relationship, so it does not include *Helio. virescens*. Therefore, there are nine moth insect species and eleven non-Noctuidae arthropods included in this part. Given that the horizontally transferred TEs could produce multiple copies in the host genome, and an HTT event occurred before the species divergence would retain copies of the horizontally transferred TE in all genomes of the diverged species, multiple HTT events could be detected in the diverged species, leading to overestimating the number of horizontal transfer events. To exclude the overestimation, the heuristic methods and clustering algorithms were applied by identification of the minimum number of HTT events for insects [15] and for vertebrates [46]. A total of 37 events were initially identified between the noctuid species and eleven non-noctuid arthropod species. After performing clustering, 37 events can be divided into three minimal events (Appendix A, represented using hitGroup), including two horizontally transferred DNA/*Helitron* that occurred between *A. pisum* and Noctuidae species and one horizontally transferred DNA/*Mariner* occurred between the *M. martensii* and the *B. fusca* (Figure 7). One *Helitron* HTT event involved 12 HTT events involving five noctuid species, such as *B. fusca, S. frugiperda*, *S. exigua*, *S. litura,* and *M. configurata*. Following the method described in 2.4, we estimated the *Helitron* element inserted into *B. fusca* genome about 71 Mya (K = 41.19%, T = k/2r). According to the phylogenetic tree, this was before the divergence of the five noctuid species. Therefore, the HTT event probably occurred in the common ancestor of these species. However, we did not detect the *Helitron* in other examined noctuid species, possibly because only limited Noctuidae species were included in the present research. Given the highly species-diverse family of Noctuidae, further research on the genomes of additional noctuid species would be necessary to determine the ancestor lineage in which this *Helitron* element was inserted. Another *Helitron* HTT event involved 19 HTT events involving three species in the Spodoptera genus. The insertion time of the *Helitron* copy into *S. frugiperda* was estimated to be 74 Mya (K = 43.01%, T = k/2r). While the *Mariner* HTT event contained six HTT events that occurred between the *M. martensii* and *B. fusca* (Appendix A). The estimated insertion time was about 44 Mya. The phylogenetic tree estimated the divergence of *B. fusca* from a common ancestor of about 49 Mya; therefore, the HTT event was highly likely species-specific.

## 4. Discussion

### 4.1. TEs Shape the Genome Diversity of Noctuidae Species

Unlike protein-coding genes under selective pressure, TE sequences are usually not subject to selective pressure and thus change rapidly [48]. In addition, TE expansion/contraction occurs at a high frequency in the genome of arthropods [13], leading to enormous variations of TEs in arthropod genomes. Therefore, it is important to understand the TE characteristics and genome-wide diversity in Noctuidae species. This study constructed a consensus sequence library for ten Noctuidae species, containing 1038–2826 TE consensus in each genome. TEs showed high variations among Noctuidae species, even among species of the same genus. The genome content of TEs also varied greatly (from 11.33% to 45.1%) among the ten species. The high variation of TE content among closely related species was consistent with previous studies on Lepidoptera species, where TEs account for 4.7–38.3% of the genomes [7], and on Insecta species, where TEs account for 1–55% of the genomes [13]. It was suggested that the increase/decrease in TE content was the most important reason affecting the genome size of arthropods [12]. Similarly, in the Noctuidae, we found a strong positive correlation between TE content and genome size (r > 0.8, *p* < 0.01). In particular, we revealed that LINE and DNA transposons contributed most to the genome sizes, unlike SINEs with no significant correlation. However, a study based on more than ten arthropods found that LINE, SINE, LTR, and DNA transposons were all positively correlated with genome sizes (r > 0.6) [11]. The discrepancy might be due to the relatively smaller content of LTR and SINE in the noctuid genomes compared to other arthropods.

Noctuid species also exhibited significant differences in the copy numbers and lineage-specific expansion of TE subfamilies. The SINE/*5S* subfamily was one of the examples whose copy number highly varied among closely related species. Copy number of the SINE/*5S* subfamily was only 130–204 copies in the three species of genus *Spodoptera,* but more than 100,000 copies in *B. fusca.* While among the ten species, *B. fusca* had the closest phylogenetic relationship to the genus *Spodoptera*. It suggested an expansion of SINE/*5S* in the genome of *B. fusca.* Activity estimation found a propagation peak of SINE/*5S* about 6 Mya in *B. fusca* (Figure 4E), suggesting elements in the SINE/*5S* are probably still active recently. Three species in the genus of *Spodoptera* allow the investigation of lineage-specific TE propagation among closely related species. We found an obvious expansion of the LTR/*Gypsy* subfamily specific in *S. exigua* but not in *S. frugiperda* and *S. litura*. The expansion event occurred very recently with a propagation peak of about 2 Mya, long after the divergence of *S. exigua* from other species (Figure 3F), and represented a species-specific expansion event.

We found that SINE/*B2* was a lineage-specific subfamily presented only in *T. ni*. Since *T. ni* diverged first from other Noctuidae species in phylogeny, the subfamily might either result from a loss of *B2* in other noctuid species or from the insertion of *B2* specific into *T. ni* through HTT. We further investigated SINE/*B2* subfamily in other genomes by comparison of *B2* consensus sequences with the RefSeq representative genomes of 272 Arthropoda species (Appendix A) using the blastn tool (default parameters) but did not find the sequence in any genome other than *T. ni*. Furthermore, we employed the NT database to search SINE/B2 sequence and found similar sequences in the five Lepidopteran species (*Selenia dentaria*, *Pseudoips prasinana*, *Herminia tarsipennalis*, *Callimorpha dominula*, and *Tyria jacobaeae*) with the hit counts more than 3. All five species were not Noctuidae species and had a far phylogenetic relationship to the Noctuidae family. The *B2* consensus sequence was from RepeatModeler based on the published TE library and was not identified by machine learning; thus, the classification was reliable. It indicated that the SINE/*B2* either was an independently evolved subfamily in *T. ni* or probably was horizontally transferred from other Lepidopteran species. The insertion was estimated from 60 Mya with a peak propagation around 20–30 Mya. However, where the *B2* subfamily came from and how it integrated into the *T. ni* genome requires further study.

### 4.2. TE Expansion Activity Correlated with Phylogeny of Noctuidae Species

In addition to being the main contribution factor to the genome size of Noctuidae species, we also investigated which class/subfamily of TE was correlated with the phylogeny of Noctuidae. Among the four classes of TE, only LINE showed a phylogenetic signal with high confidence, indicating the essentially vertical inheritance characteristics of LINE elements in Noctuidae. In particular, four LINE subfamilies, *CR1*, *L2*, *RTE*, and *R1,* showed a high correlation with Noctuidae phylogeny, all abundant in copy number. In contrast, despite the high copy number of DNA/*Helitron* subfamily in noctuid genomes, its correlation with phylogeny was insignificant. This was probably because of the different integration mechanisms of the *Helitron*. Another potential reason is that elements in the *Helitron* subfamily had been involved in HTT events in the Noctuidae species, which we will discuss below.

We further elucidated whether the expansion of the TE class/subfamily contributed to the evolution of noctuid genomes. We noted that the LINE, LTR, and DNA transposons all had relatively low activity in the genus *Helicoverpa,* that was probably why both species in the genus *Helicoverpa* had the least TE content and the smallest genome sizes. In contrast, the LINE, LTR, and DNA transposons all expanded in the genome of *B. fusca* (Figure 6), accumulating the highest TE content of *B. fusca* (45.1%) in the ten species. By calculating the change rate of TE, interestingly, we found only LINE and DNA transposons had their expansion occurred very recently (Figure 4), indicating these TEs were highly likely active in *B. fusca*, especially the LINE/*CR1*, LINE/*R1*, and LINE/*RTE* subfamily who showed recent expansion in *B. fusca*.

Although LINE and DNA transposons largely impacted the genomes of Noctuidae species, this was not the case in the recent evolution of a specific species. For example, the activity of the TE class/subfamily in *S. exigua* was substantially different from the other two *Spodoptera* species. LTR elements expanded in the *S. exigua* genome, and the expansion occurred very recently (Figure 3F), accumulating the highest genome content of LTR (4.22%) among the ten species, in contrast to a reduction in LTR in the *S. frugiperda* and *S. litura* genomes. We identified several HTT events related to LTR elements in the *S. exigua* genome (discussed below), which may contribute to its recent expansion. Thus, LTR is the TE class that had the most important impact on the recent evolution of *S. exigua*.

### 4.3. HTT Events on the Genomes of the Noctuid Moths

Despite the essential homoplasy-free characteristics of TE, HTT events have been widely reported in the insect genomes [17]. Peccoud’s study found that as high as 2248 HTT events occurred among 195 insect species in the last 10 million years, which probably was only a tiny fraction of the actual HTT events between insects [45]. Another study analyzed 460 species of arthropods for HTT and found significantly more HTT events in Lepidoptera than in other arthropods [19]. Our study identified 56 possible HTT events among the ten noctuid genomes. Previous studies indicated that the higher the copy number of a subfamily, the higher the probability that its member was misclassified as an HTT event [49]. By calculating the frequency of HTT events per thousand TE copies, we did not observe the trend in our results, suggesting our results do not suffer from the copy number bias. A large-scale study on the HTT in insect genomes identified 2248 HTT total events, 1087 of which were associated with DNA/*Tc1-Mariner* subfamily [45]. In our study, 17 of the 56 HTTs belonged to elements in the *Tc1-Mariner* subfamily, which was the highest among all subfamilies. The *Tc1-Mariner* is short in length (1–2 Kb), which may facilitate HTT through the vector resulting in a high frequency of HTT events.

One HTT event involved an LTR/*Gypsy* element transfer between *S. exigua* and *M. configurata*. By calculating the divergence of the *Gypsy* consensus of *S. exigua*, we estimated the divergence rate of the *Gypsy* consensus was 1.38% in the *S. exigua*, converting to an insertion time of 2.37 Mya. It was consistent with the activity estimation of *Gypsy* with a burst propagation around 2 Mya in the *S. exigua* genome. When TEs inserted into a new genome by HTT, the new host genome was generally unable to immediately inhibit the replication and translocation of the transposons. If the TE maintained replication capacity in the new genome, they might lead to a massive TE replication [15]. Therefore, we inferred that the horizontally transferred *Gypsy* element could have led to the recent mass replication of *Gypsy* in the *S. exigua* genome, accounting for 2.25% of the entire genome, about 4.5–6.4 times that in the closely related species of the same genus.

It is noteworthy that for the HTT events identified between closely related species, no matter how strict the threshold, there is still the possibility that the similarity of vertically inherited TEs exceeds the selection threshold. Previous studies suggested that the closer the distribution and the closer the species’ affinity, the higher the frequency of HTT occurring between species [45]. The study further suggested a minimal number of HTT events to identify HTT events between species that diverged more than 40 Mya to avoid exaggerating the number of HTT events. Following the method, we identified a minimum of three independent HTT events between the noctuid moths and eleven non-noctuid arthropods. These HTT events occurred in the genome of five noctuid species, including *B. fusca*, *M. configurata*, and three *Spodoptera* species. Previous studies suggested that genomes with more HTT events probably had higher TE contents and larger genome sizes [11]. All five noctuids showed relatively larger genome sizes and higher TE contents compared to the other five species, indicating HTT events might have shaped the evolution of Noctuidae genomes by leading to TE expansion.

Among the three minimum number HTT events, one *Mariner*–related HTT occurred specifically in *B. fusca.* We further searched the *Mariner* sequence of *B. fusca* in other arthropod genomes. We found a highly similar sequence from *Cyphomyrmex costatus* in Hymenoptera, supporting the HTT event of this element between different insect genomes. However, whether the *Mariner* element transferred from *M. martensii* to the *B. fusca* genome needs further study with more arthropod genomes.

## 5. Conclusions

In conclusion, this study constructed a consensus sequence library for ten Noctuidae species based on multiple methods, significantly improving TE annotation in the Noctuidae genomes. By comparison of the TE genome content, TE composition, and propagation activity of TE class/subfamilies among the ten Noctuidae species, this study provided new insights into the essential contributions of TEs to the genome size variation, genomic diversity, and phylogeny of Noctuidae species. We identified lineage-specific TE subfamilies and recent expansion of TE subfamilies in some Noctuidae species, suggesting they were probably still active in the Noctuidae genomes. Moreover, 56 potential HTT events were identified among the noctuid species, and a minimum of 3 independent HTT events between the Noctuidae species and 11 non-noctuid arthropod species. The HTT events could account for the recent expansion of *Gypsy* subfamily in the *S. exigua* genome and the species-specific expansion of *Mariner* subfamily in *B. fusca*.

## Figures and Tables

**Figure 1 genes-14-01244-f001:**
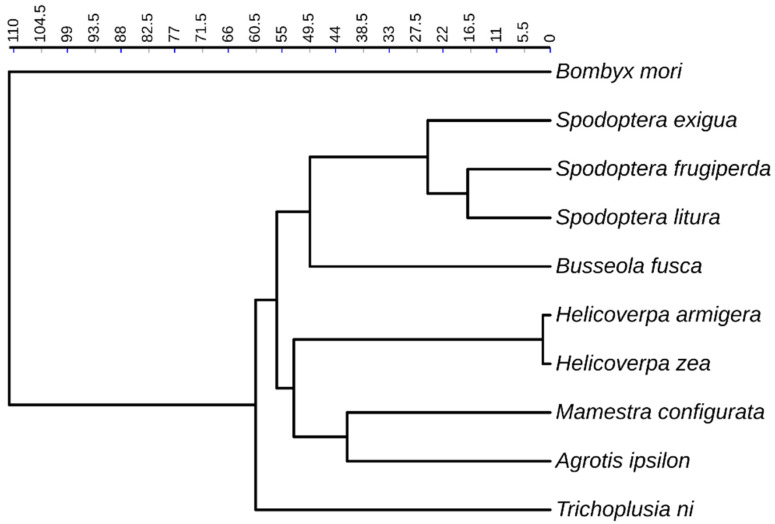
Phylogenetic relationships among the nine noctuid moth species with the silkworm (*Bombyx mori*) as the outgroup. *Helio. virescens* is not included in the tree because of its unknown position.

**Figure 2 genes-14-01244-f002:**
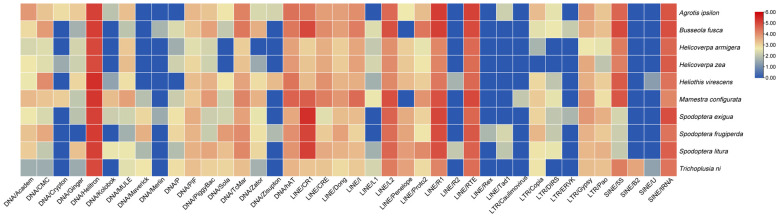
TE families/subfamilies and their copy numbers of the ten noctuid genomes. Each value was log10 transformed.

**Figure 3 genes-14-01244-f003:**
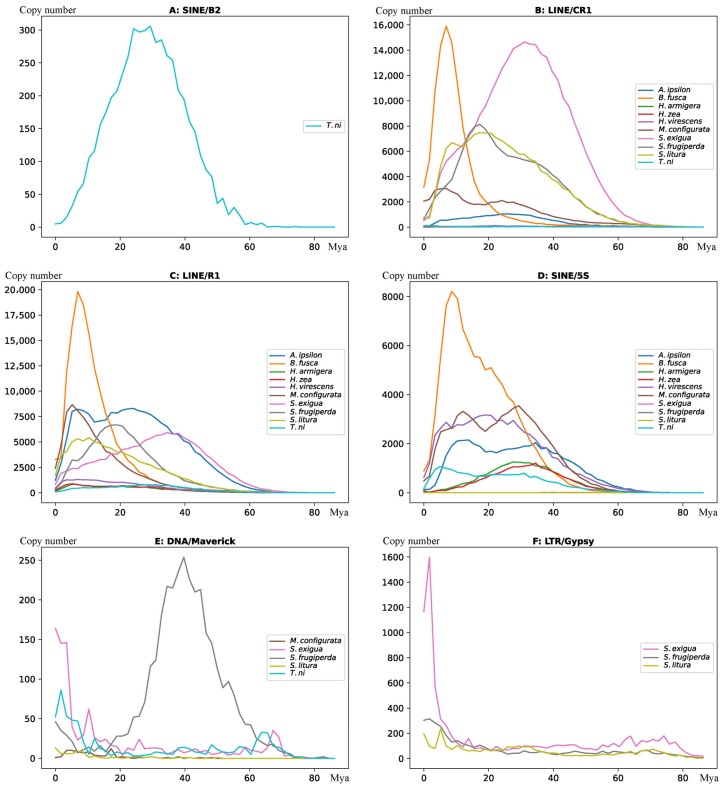
Illustration of activity of different subfamilies of TEs.

**Figure 4 genes-14-01244-f004:**
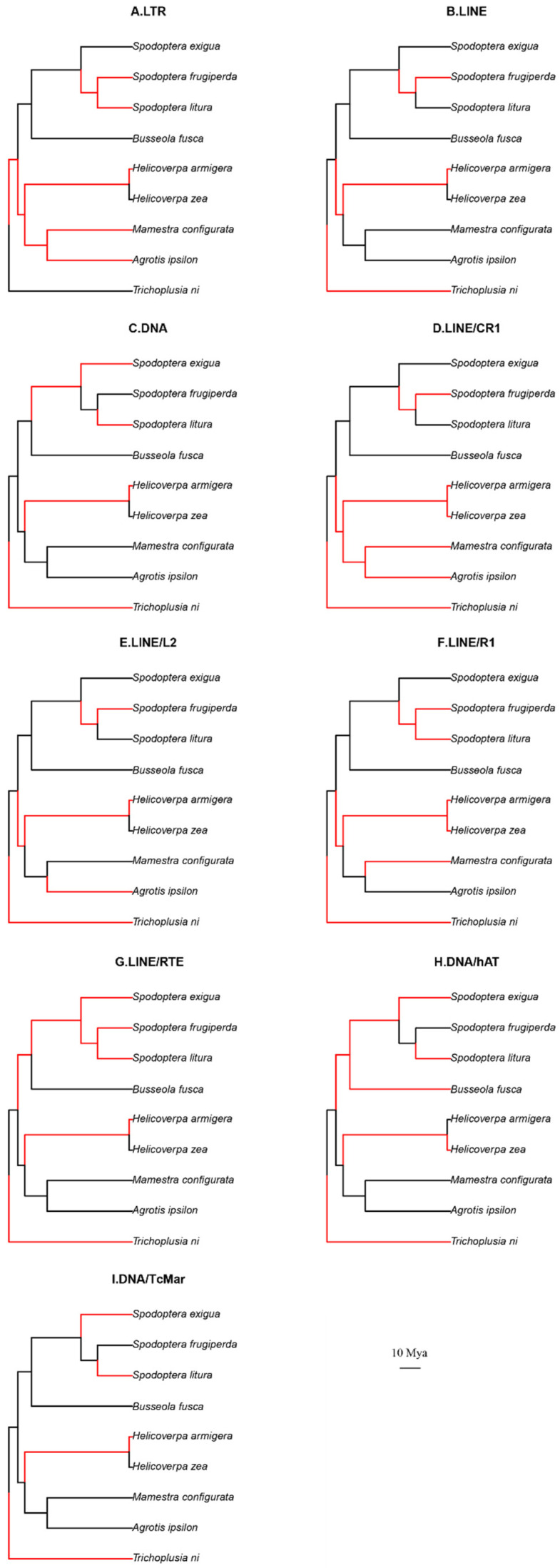
The expansion activity of TEs in nine noctuids as estimated by TE load. The black and red lines on the phylogenetic tree represented expanded or reduced TE load in the genomes, respectively. (**A**): LTR retrotransposon, (**B**): LINE, (**C**): DNA transposon, (**D**): LINE/CR1, (**E**): LINE/L2, (**F**): LINE/R1, (**G**): LINE/RTE, (**H**): DNA/hAT, (**I**): DNA/TcMar.

**Figure 5 genes-14-01244-f005:**
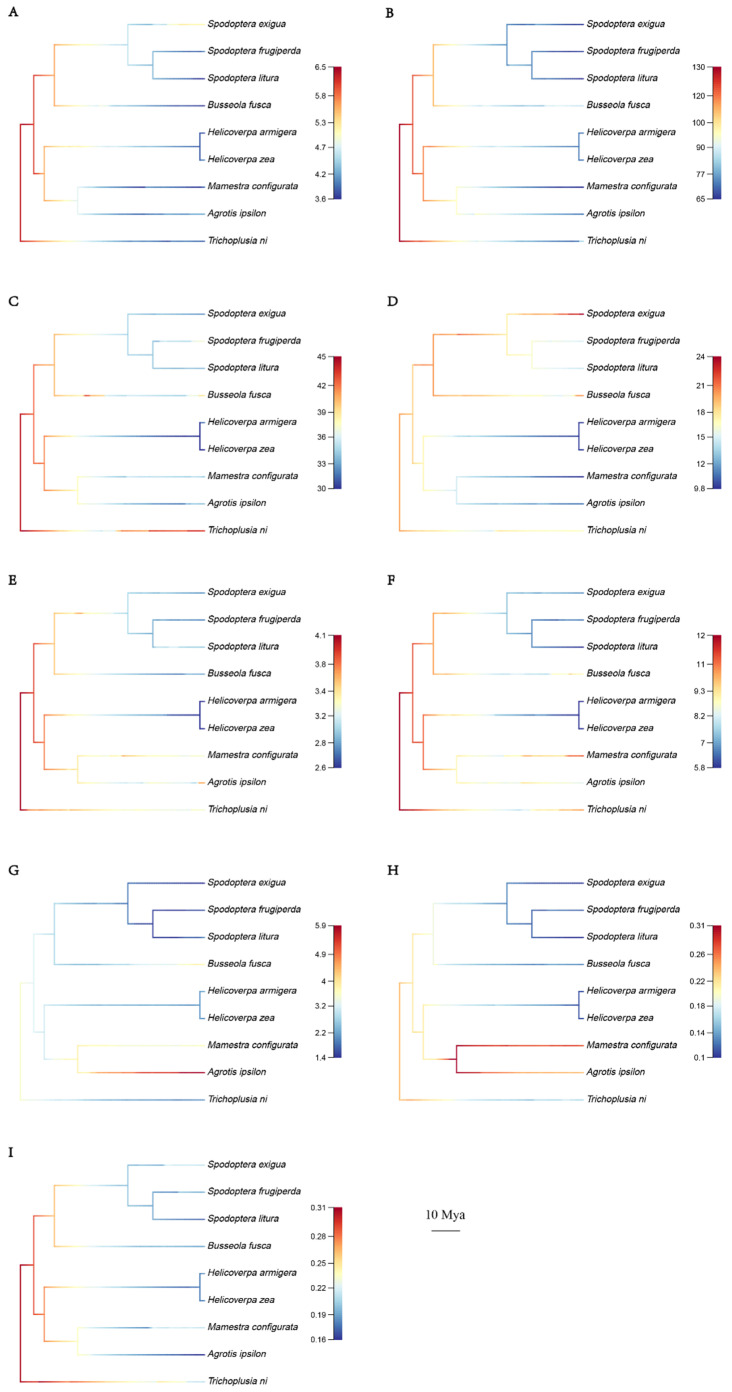
The dynamic change rates of TE class/subfamilies in the phylogeny of Noctuidae species. The TE load’s evolution rate is used to determine the color of the branch, with the rate increasing from cool (blue) to warm (red). (**A**): LTR retrotransposon, (**B**): LINE, (**C**): DNA transposon, (**D**): LINE/CR1, (**E**): LINE/L2, (**F**): LINE/R1, (**G**): LINE/RTE, (**H**): DNA/hAT, (**I**): DNA/TcMar.

**Figure 6 genes-14-01244-f006:**
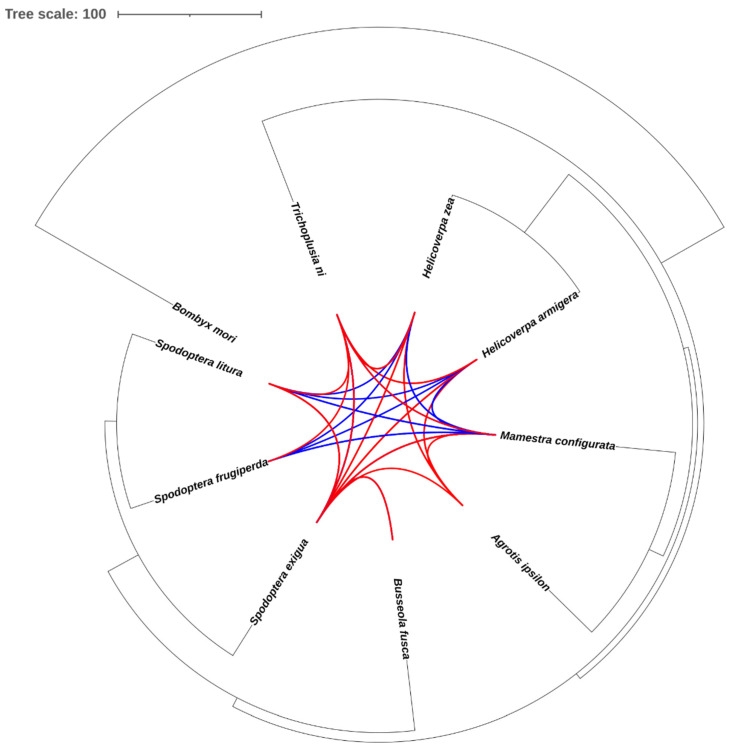
The potential HTT events among the noctuid species. HTT events from RNA transposons are represented by red lines, while DNA transposons are represented by blue lines.

**Figure 7 genes-14-01244-f007:**
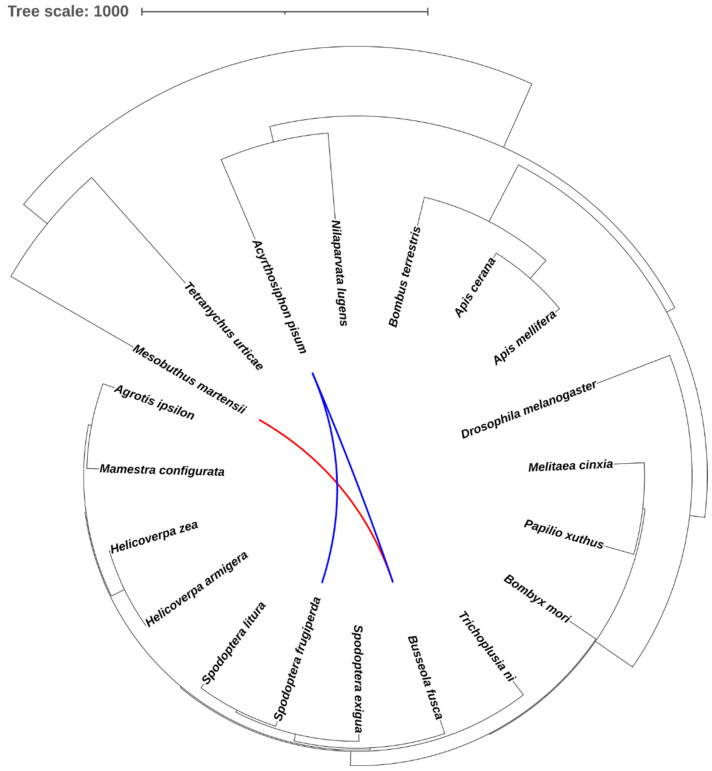
The minimum number of HTT events between the 9 noctuid species and eleven non-noctuid arthropod species. HTT events from DNA/*Helitron* are represented by blue lines, while red lines represent DNA/Mariner.

**Table 1 genes-14-01244-t001:** Consensus sequences of transposable elements in the ten noctuid species.

Level	Species Name	Genome Size (Mb)	RTE	DTE	UnC	Total
Scaffold	*Agrotis ipsilon*	486.92	1238	1120	131	2489
Scaffold	*Busseola fusca*	490.17	1617	1067	136	2820
Scaffold	*Helicoverpa armigera*	299.98	562	607	52	1221
Scaffold	*Helicoverpa zea*	306.41	638	637	67	1342
Scaffold	*Heliothis virescens*	403.15	790	855	85	1730
Scaffold	*Mamestra configurata*	559.39	1509	1186	131	2826
Chromosome	*Spodoptera exigua*	446.8	831	456	43	1330
Chromosome	*Spodoptera frugiperda*	486.23	743	644	63	1450
Chromosome	*Spodoptera litura*	428.03	1006	556	70	1632
Chromosome	*Trichoplusia ni*	367.26	578	428	32	1038

Retrotransposable elements, DNA transposons, and unclassified TEs are represented by RTE, DTE, and UnC, respectively.

**Table 2 genes-14-01244-t002:** Transposable element loads and genome size in the ten noctuid species.

Species	DTE (%)	LTR (%)	LINE (%)	SINE (%)	UnC (%)	All (%)	Genome Size (Mb)
*Trichoplusia ni*	4.59	2.63	4.99	2.25	1.4	15.86	367.2
*Helicoverpa armigera*	4.01	0.74	3.46	2.06	1.06	11.33	299.98
*Helicoverpa zea*	4.82	1.07	4.05	2.63	1.19	13.76	306.41
*Spodoptera exigua*	4.74	4.22	17.64	2.56	1.48	30.64	446.80
*Spodoptera litura*	5.87	2.4	14.81	3.39	2.51	28.98	428.03
*Spodoptera frugiperda*	9.12	1.94	12.54	0.98	2.17	26.75	486.23
*Agrotis ipsilon*	11.8	2.08	13.81	3.16	3.42	34.27	486.92
*Mamestra configurata*	11.24	2.12	15.59	2.37	3.4	34.72	559.39
*Busseola fusca*	12.1	3.16	20.16	6.11	3.57	45.10	490.17
*Heliothis virescens*	8.39	1.3	5.64	2.92	2.29	20.54	403.15

DNA transposons and unclassified TEs are represented by DTE and UnC, respectively.

**Table 3 genes-14-01244-t003:** The correlation coefficient between the genome sizes and TE loads.

Classes of TEs	Subfamily	r Value	*p*-Value
**DNA transposon**		**0.81**	**4.21 × 10^−3^**
	DNA/CMC	0.31	0.4
	**DNA/TcMar**	**0.97**	**2.17 × 10^−6^**
	**DNA/Zator**	**0.74**	**0.01**
	DNA/hAT	0.46	0.18
	DNA/PiggyBac	0.21	0.55
**LINE**		**0.83**	**3.2 × 10^−3^**
	**LINE/Dong**	**0.91**	**2.56 × 10^−4^**
	**LINE/L2**	**0.77**	**9.6 × 10^−3^**
	**LINE/R1**	**0.75**	**0.01**
	LINE/RTE	0.58	0.08
	LINE/Jockey	0.54	0.1
	LINE/Proto2	0.44	0.21
	LINE/CR1	0.41	0.24
	LINE/CRE	0.34	0.34
	LINE/Rex	0.25	0.49
LTR		0.49	0.15
	LTR/Copia	0.4	0.26
SINE		0.2	0.57
	SINE/tRNA	0.2	0.58

The bold line in the table indicated TE classes with significant *p*-values.

**Table 4 genes-14-01244-t004:** Lambda value and K value of the TE load.

Classes of TEs	Subfamily	Lambda Value	*p*-Value (Lambda)	K Value	*p*-Value (K)
DNA		0.93	0.23	0.75	0.16
	**DNA/hAT**	**1.02**	**7.1 × 10^−3^**	**1.13**	**0.01**
	DNA/Helitron	0.89	0.72	0.6	0.22
	DNA/MULE	0.84	0.22	0.66	0.16
	**DNA/TcMar**	**0.97**	**0.03**	**1.13**	**0.03**
**LINE**		**0.99**	**0.01**	**1.33**	**0.02**
	**LINE/CR1**	**1.01**	**5.7 × 10^−3^**	**1.62**	**4 × 10^−3^**
	LINE/Dong	0.99	0.07	0.99	0.02
	LINE/Jockey	0.73	0.32	0.52	0.24
	**LINE/L2**	**0.99**	**0.03**	**1.19**	**0.03**
	**LINE/R1**	**1.02**	**6.6 × 10^−3^**	**1.49**	**0.01**
	**LINE/RTE**	**1.01**	**0.03**	**1.18**	**0.03**
LTR		0.79	0.12	0.57	0.24
	LTR/Pao	0.70	0.17	0.39	0.33
SINE		6.8 × 10^−5^	1	0.42	0.43
	SINE/tRNA	1.01	0.21	0.82	0.1

The bold line in the table indicated TE classes with significant *p*-values.

**Table 5 genes-14-01244-t005:** Range of change rates of transposons.

Transposon Type	Range of Change Rates
LTR	3.6~6.5
LINE	65~130
LINE/CR1	9.8~24
LINE/L2	2.6~4.1
LINE/R1	5.8~12
LINE/RTE	1.4~5.9
DNA Transposon	30~45
DNA/hAT	0.1~0.31
DNA/TcMar	0.16~0.31

**Table 6 genes-14-01244-t006:** The number of HTTs in each TE subfamily.

Subfamily	Frequency of HTTs in Every Thousand TE Copies	Number of HTTs	Copy Number of TEs
LTR/Gypsy	0.04	1	22,547
LTR/Copia	0.25	1	3978
LINE/RTE-RTE	0.05	10	183,155
LINE/RTE-BovB	0.03	4	130,583
LINE/R1	6.1 × 10^−3^	3	485,681
LINE/Proto2	0.05	1	21,637
LINE/L2	0.03	6	209,382
LINE/Dong-R4	0.13	3	22,186
LINE/CR1-Zenon	2.6 × 10^−3^	1	380,795
LINE/CR1	0.25	4	15,874
DNA/Zator	0.18	1	5418
DNA/TcMar-Tc1	0.44	12	27,452
DNA/TcMar-Mariner	0.38	5	13,007
DNA/Helitron	1.2 × 10^−3^	1	774,822
DNA/Maverick	1.72	3	1741

## Data Availability

The data supporting the findings of this study are openly available from the NCBI (https://www.ncbi.nlm.nih.gov/) (accessed on 30 September 2020) and ArTEdb (http://artedb.net/) (accessed on 30 September 2020). Accession numbers are listed in Appendix A.

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
