# Peer review of "Transposable Elements Shape the Genome Diversity and the Evolution of Noctuidae Species"

_genes, 2023, doi:10.3390/genes14061244_

Round 1

Reviewer 1 Report

Overall comment:

This study provides helpful information to insect TE researchers and is considered to have conducted a comprehensive and labor-intensive study using bioinformatics.

The study annotated and characterized transposable elements (TEs) in 10 Noctuidae species, revealing high variation in TE content, with a positive correlation between TE content and genome size. The expansion of TEs, especially LINEs, played a role in the evolution of Noctuidae genomes, and horizontal transfer TE (HTT) events were identified, indicating substantial impacts on genome evolution. Species-specific subfamily expansions and a recent Gypsy subfamily expansion in S. exigua were also identified.

However, the authors should revise the minor points that I addressed below.

Minor points

1. There are many spacing problems between the genus and the specific name. It may have occurred during the PDF conversion process, so please check carefully.

I found lines 24, 99, 104, 214, 223, 224, 276, 277, 356, 383, 433, 535, 536, 539, 544, 564, 576, but you can find more errors.

2. There is a space error between periods or commas after parentheses.

Lines 111, 117, 124

3. There is a common spacing error.

Lines 150, 229, 310, 335, 346, 352, 477, 479, 484, 492, 504, 539

4. Replace “A, B, C or A, B and C” with “A, B, and C”.

Lines 311, 322, 330, 336, 354, 492, 503

5. Line 38: Replace “jump” with “mobilize”.

6. Line 56: Replace “transposons the predominant” with “transposons are the predominant”. The verb is missing.

7. Line 71: Replace “Drosophila melanogaster” with “D. melanogaster”.

8. Line 90 ~92: Species name should be italicized.

9. Figure 2 has been cropped. The whole picture is not visible.

10. There is no unit for the x-axis and y-axis of figure 3. Please add the unit.

11. A distance scale bar should be added to figures 4 and 5.

12. The species names in Figures 6 and 7 are too small to read.

Reviewer 2 Report

In this work the authors have constructed TE consensus sequence libraries for ten Noctuidae species using multiple annotation pipelines, improving TE annotation in these genomes. This study provided insights into the contributions of TEs to the genome size and genomic diversity by comparison of the TE composition and propagation activity, correlating it also with the phylogeny of Noctuidae species. By inspecting the TE dynamics and HTT events, this study illuminated the impact of TE activities on the Noctuidae genome evolution.

I consider the scientific topic addressed to be interesting. This work presents a nice contribution to the research area of repetitive DNA sequences, which is enriched by obtaining information from different model systems and diverse eukaryotic genomes. The methodology employed is versatile and appropriate, and the manuscript is well written.

However, prior to the publication, certain parts of the manuscript need clarification. My questions, comments and suggestions for the authors are listed below:

Line 23: I suggest replacing “HTT events caused by a Gypsy transposon” with “HTT events of a Gypsy transposon”, as the current form would imply that Gypsy is only a vector in HTT.

DNA/Helitrons are DNA transposons which are proposed to transpose by a mechanism similar to rolling-circle (RC) replication, hence also being called DNA/RC. The authors alternate these two terms (DNA/Helitrons and DNA/RC) throughout the manuscript, tables and figures. This becomes more pronounced in the HTT section when among Noctuidae species DNA/RC had the lowest frequency of HTT per copy with the highest copy number among all subfamilies, and DNA/Helitron seems to be actively included in HTT between Noctuidae species and other arthropods. In case authors consider these two terms (DNA/Helitrons and DNA/RC) to be synonyms, please chose one and unify the name throughout the manuscript. In case they consider DNA/Helitrons only a subset of DNA transposons employing RC, please state what else DNA/RC term includes.  

Suppl. Table 6: explanation should be provided what column titles present, especially terms: q, s, ka, ks, community.

Lines 424-425: Please indicate from which results it is visible that Helitron element inserted into B. fusca genome about 71 Mya.

Lines 426-427: If the authors propose that the HTT event probably occurred in the common ancestor of these species, why this Helitron was not found in the genomes of other inspected Noctuids, as would be expected? Please clarify this part.

428-429: Please indicate from which results it is visible that Helitron element inserted into S. frugiperda genome 74 Mya.

Lines 429-430: Please explain how HTT event was genus-specific only for Spodoptera if insertion time of 74 Mya precedes the divergence time of all inspected Noctuids? If the authors propose that the HTT event probably occurred in the common ancestor of Spodoptera, why this Helitron was not found in the genomes of other inspected Noctuids, as 74 Mya was in the time of the common ancestor, before their divergence?

Line 478-481: The search could be expanded simply employing online NCBI blast tool and blasting SINE/B2 sequence against WGS or RefSeq Genome data of all available insect genomes (and wider).

The authors should make publicly available the TE consensus sequence libraries obtained in this work, in ArTEdb or elsewhere.

minor observations:

Line 44-45: I suggest introducing original references for genome sizes of those two species.

Line 56: “Melpomene” should be all lowercase.

Line 65, 66, etc: the usage of the full term “horizontal transfer” is redundant as abbreviation HTT has been introduced in line 63.

Line 66-67: I suggest exchanging “from an old host to a new one” with “from one host to another”.

Lines 90-93: Names of the species and genera (Helicoverpa, Spodoptera) should be in italic.

Line 231: Please correct “LTR transposons” into “LTR retrotransposons”.

Line 240: TE subfamilies are presented in the Figure, not families, as stated in the figure description.

Figure 3: Labels for the x and y axes are missing from the plots.

Round 2

Reviewer 2 Report

The authors responded adequately to the comments and suggestions, therefore I recommend the publication of their work in Genes.
Congratulations to the authors.